# Light-Photoreceptors and Proteins Related to *Monilinia laxa* Photoresponses

**DOI:** 10.3390/jof7010032

**Published:** 2021-01-07

**Authors:** Silvia Rodríguez-Pires, Eduardo A. Espeso, Neringa Rasiukevičiūtė, Paloma Melgarejo, Antonieta De Cal

**Affiliations:** 1Department of Plant Protection, Instituto Nacional de Investigación y Tecnología Agraria y Alimentaria (INIA), Ctra. de La Coruña Km. 7, 28040 Madrid, Spain; silvia.rguezpires@gmail.com; 2Department of Cellular and Molecular Biology, Centro de Investigaciones Biológicas (CIB), Consejo Superior de Investigaciones Científicas, Ramiro de Maeztu 9, 28040 Madrid, Spain; eespeso@cib.csic.es; 3Laboratory of Plant Protection, Lithuanian Research Centre for Agriculture and Forestry, Institute of Horticulture, Kaunas St. 30, LT-54333 Babtai, Kaunas, Lithuania; neringa.rasiukeviciute@lammc.lt; 4Dirección General de Producciones y Mercados Agrarios, Ministerio de Agricultura, Pesca y Alimentación, 28071 Madrid, Spain; pmelgarejo@mapa.es

**Keywords:** photoreceptors, light response, *Monilinia laxa*, brown rot, photobiology

## Abstract

Light represents a ubiquitous source of information for organisms to evaluate their environment. The influence of light on colony growth and conidiation was determined for three *Monilinia laxa* isolates. The highest mycelial growth rate was observed under red light for the three *M. laxa* isolates, followed by green light, daylight or darkness. However, reduced sporulation levels were observed in darkness and red light, but conidiation enhancement was found under daylight, black and green light with more hours of exposure to light. Putative photoreceptors for blue (white-collar and cryptochromes), green (opsins), and red light (phytochromes) were identified, and the photoresponse-related regulatory family of velvet proteins. A unique ortholog for each photoreceptor was found, and their respective domain architecture was highly conserved. Transcriptional analyses of uncovered sets of genes were performed under daylight or specific color light, and both in time course illumination, finding light-dependent triggered gene expression of *MlVEL2*, *MlPHY2*, *MlOPS2*, and *MlCRY2*, and color light as a positive inductor of *MlVEL3*, *MlVEL4*, *MlPHY1*, and *MlCRY1* expression. *M. laxa* has a highly conserved set of photoreceptors with other light-responsive fungi. Our phenotypic analyses and the existence of this light-sensing machinery suggest transcriptional regulatory systems dedicated to modulating the development and dispersion of this pathogen.

## 1. Introduction

The European brown rot *Monilinia laxa* (Aderhold et Rulhand) Honey causes diseases in important *Rosaceae* family crops, in particular, stone fruit and pome fruit [1]. Climatic conditions are critical for *M. laxa* infection in stone fruit. Among the climatic factors that most influence the penetration and infection of *Monilinia* spp. are temperature and wetness duration period [2]. A positive correlation has been reported between *M. laxa* incidence at postharvest and temperature, temperature being responsible for 82% of postharvest brown rot [3]. The temperature was also positively correlated with the number of necrotic infected twigs in peach orchards [4]. Survival, colonization, latency, reproduction, release, transport, and deposition of *Monilinia* spp. conidia are related to temperature, the relative humidity, the amount of rainfall, and the wind direction [5,6]. 

Other less studied abiotic environmental factors, such as light and circadian rhythms, are important sources of information for organisms to evaluate their environment. Light has been recognized as one of the environmental factors stimulating pathogens for host invasion [7]. Interest in plant–pathogen interactions has increased because of its possible role in the development of disease [8,9], photoreceptors being even necessary for complete virulence [10,11,12]. Light responsive *Botrytis cinerea* strains show different colonial growth behaviors through sensing specific ranges of light wavelength. The ultraviolet (UV) and far-red lights promote conidiation, while red and blue lights have a negative effect [13]. The effect of green light was less studied in *B. cinerea*, although it has been related to repression of mycelial growth and conidial germination [14]. In *Monilinia* spp. long-wave UV light is a useful tool to distinguish between species; a faster growth rate was observed in *M. fructigena* through light–dark cycles or UV light [15]. Light triggers spore production in the field through diurnal patterns of light–darkness [5]. Brown rot caused by *M. laxa* is related to the intensity and quality of light as well as the light duration period. An increase in conidia density on nectarines was recorded with greater exposure to light during *M. laxa*–fruit interaction [16]. Red light also causes a significant increase in the incidence and severity of nectarine brown rot caused by *M. laxa* [16]. 

Fungi have the ability to sense light through photoreceptors that can respond to direction, different wavelengths, or intensity of light [9,17,18]. As part of the signaling networks, the activity of photoreceptors influences multiple biological responses, such as spore germination, vegetative growth, sexual and asexual development, secondary metabolism, and circadian rhythm [19,20,21]. Photobiology has been thoroughly examined in fungal models, such as *Neurospora crassa* and *Aspergillus nidulans*, harboring similar genetic components but subject to differential regulation. For example, the white-collar complex is the central photosensory system in *N. crassa,* while multiple photoreceptors are needed in *A. nidulans* [22,23,24]. Light is essential for conidiation in *A. nidulans* regulated by the *veA* gene product (velvet family), both asexual development and secondary metabolism [25,26,27]. Members of the velvet family are probably one of the most ancient fungal regulators connecting signal transduction with the control of development and metabolism [28]. The continuous increase in information on fungal genomes allows wide-ranging searches for candidate photoreceptors that identify highly conserved groups; white-collar and cryptochrome families as blue light receptors, phytochromes, and opsins for the red light and green light-sensing, respectively [9,20,29]. 

The closely related *B. cinerea*, *Sclerotiniaceae* family member like *Monilinia* spp., possesses a remarkable collection of photoreceptor genes covering UV to far-red light [30]. The importance of sensing light for this pathogen became evident when a natural variation in one of the velvet genes was found and explained the light-dependent conidiation in B05.10 in contrast to the T4 isolate [11]. *M. laxa* is a light-responsive fungus with the ability to react to light [16]. The presence of light changes their gene expression patterns [31], but the machinery by which they perceive light has been scarcely explored.

The main objective of this work was to evaluate the effect of light on colony growth and conidiation of *Monilinia laxa* in vitro conditions and describe the putative photoreceptor machinery in *M. laxa* genome by in silico analyses. The transcriptional pattern of some potential photoreceptor genes in response to different light wavelengths was also determined.

## 2. Materials and Methods

### 2.1. Monilinia laxa Strains and Culture Conditions

Three single-spore strains from *M. laxa* (namely 5L, 8L, and 25L) were used in this work from the culture collection of the Plant Protection Department of INIA (Madrid, Spain). Strains were originally isolated from mummified plum fruit (cv. Sungold) from a commercial orchard in Lagunilla (Salamanca, Spain). The genome sequence of *M. laxa* 8L was recently published [32], and the strain was also deposited in the Spanish Culture Type Collection (Spanish acronym, CECT 21100). *M. laxa* strains were stored at 4 °C on potato dextrose agar for short-term storage and −80 °C as a conidial suspension in 20% glycerol for long-term storage. Cultures were incubated at 22 °C with a 12 h photoperiod for conidiation on potato dextrose agar amended with 20% of tomato pulp (PDA-T; Difco, Detroit, MI, USA) for 7–9 days.

### 2.2. Effect of Light on Monilinia laxa Colony Growth and Conidiation

The influence of different illumination conditions on colony growth rate and conidia production under two photoperiods was evaluated for the three *M. laxa* isolates described above. A 15 µL droplet of conidia suspension of each strain (10^5^ conidia ml^−1^) was placed on the center of Petri dishes filled with PDA-T and incubated for 7 days at 22 °C in the indicated light environment. To determine the effect of light on *M. laxa* growth, two perpendicular diameters were measured daily from 3-day to 7-day incubation. The daily growth length (diameter in mm day^−1^) was calculated from the individual measurements of the colony diameter using regression analysis. Sporulation concerning the growth area from each strain and condition was calculated at the end of each assay. For that purpose, the surface of each plate was individually scratched with a sterile loop after adding 5 mL of sterilized distilled water and filtered through glass wool to remove the mycelia. The number of conidia produced by *M. laxa* strains in each light condition and photoperiod was counted using a hemocytometer. Data are expressed as total conidia divided by the colony area (cm^2^). Two different photoperiods were tested, 16 h light/8 h darkness or 8 h light/16 h darkness, and three replicates were made of each light condition, and the complete experiment assay was done twice. Plates belonging to continuous darkness condition were protected from light but subjected to both photoperiods. White light was generated using a set of 4 fluorescent bulbs of Osram 36 W/954 (daylight) and Osram 36 W/840 (cool light). Red light, blue light, green light, and black light (UV-A) were generated by using a set of 4 fluorescent bulbs of F36W/T8/Red Sylvania, F36/T8/B Sylvania, F36W/T8/G Sylvania, and TL-D Philips 36W fluorescent bulbs, respectively.

### 2.3. Identification of Putative Photoreceptors

To find putative photoreceptors of *M. laxa*, we used protein sequences of described photoreceptors in *B. cinerea* B05.10 (Table 1) as query sequences against the *M. laxa* 8L predicted proteome [32] using BLASTP tool through NCBI Genome Workbench software (https://www.ncbi.nlm.nih.gov/tools/gbench/). Moreover, all the sequences were verified in a recently Mlax316 *M. laxa* available genome [33], and candidates were subjected to gene ontology classification using OmicsBox v1.3.11 (BioBam, Valencia, Spain). Candidate *M. laxa* 8L proteins were further analyzed for their domain structure with InterPro Scan, Pfam, Prosite software tools using default settings and NCBI conserved domain database.

We used the Mlax316 *M. laxa* genome database [33] to design specific primers for the amplification and sequencing of the complete *velvet4* in our *M. laxa* isolates due to not appearing in our proteome prediction from the deposited genome. The presence of *velvet4* in the genomic DNA was confirmed by PCR (Appendix A). PCR products were purified using Nucleospin gel and PCR clean-up kit (MN, Düren, Germany) and sequenced (STAB VIDA, Caparica, Portugal).

### 2.4. Light Transcriptomic Analyses

Freshly harvested conidia from *M. laxa* 8L were used to inoculate potato dextrose broth (PDB; Difco, Detroit, MI, USA) with a final concentration of 10^5^ conidia ml^−1^ and incubated at 22 °C in darkness at 150 rpm for 24 h. Mycelium was collected from 50 mL of PDB by brief vacuum filtration into 0.45 µm cellulose–nitrate membrane placed on PDA-T under darkness, daylight, blue, green, and red light exposure. After 0.5, 2, 4, and 6 h of the indicated light environment, mycelium was harvested and immediately frozen in liquid nitrogen. The complete assay was repeated twice.

Total RNA was isolated from mycelia of each light and time exposure with TRI reagent (Sigma–Aldrich, St Louis, MO, USA), according to [34]. RNA concentration and purity were assessed using a Nanodrop 2000 spectrophotometer (Thermo Scientific, Wilmington, DE, USA). RNA integrity was checked by 1.2% agarose gel electrophoresis. RNA was treated with DNase I Amplification Grade (Invitrogen, Carlsbad, CA, USA) according to the manufacturer’s specifications to remove any remaining genomic DNA. cDNA was synthesized from 2 µg of RNA using the SuperScript First-Strand Synthesis System for RT-PCR and oligo(dT) primer (Invitrogen, Carlsbad, CA, USA). Real-time PCR was performed with a 7500 Fast Real-Time PCR (Applied Biosystems, Foster City, CA, USA) using GoTaq qPCR Master Mix (Promega, Madison, WI, USA). Each reaction was carried out in triplicates in a total volume of 20 µL, containing 10 µL of 2× GoTaq qPCR Master Mix, 7.8 µL of Nuclease-Free water, 0.6 µL of each primer (10 µM), and 1 µL of cDNA. The cycling program was 2 min at 95 °C, followed by 40 cycles of 15 sec at 95 °C and 1 min at 60 °C. After the amplification reaction, a melt curve analysis was performed to check the specificity of different reaction products. The relative expression levels were calculated using the 2^−ΔΔCT^ method [35] relative to darkness condition at 0 h post illumination. The *M. laxa* histone H3 (*Ml_HistoneH3*, BK012065) and 60S ribosomal protein L5 (*MlRPL5*, BK013068) were used as endogenous reference genes. All the primers used in RT-qPCR were designed base on *M. laxa* 8L genome using vector NTI (ThermoFisher, Waltham, MA, USA) and listed in Appendix A. Three technical replicates were analyzed for each biological replicate for both the target and the reference genes. Nucleotide sequence data reported are available in the Third Party Annotation Section of the DDBJ/ENA/GenBank databases under the accession numbers TPA: *Ml_HistoneH3*, BK012065; *MlRPL5*, BK013068; and BK014380 to BK014395 for putative photoreceptors.

### 2.5. Statistical Analysis

Data were analyzed by a multiway analysis of variance [36]. When F-test was significant at *p* ≤ 0.05, the means were compared using Duncan’s multiple range test.

## 3. Results

### 3.1. Effect of Light on Monilinia Laxa Colony Growth, Conidiation, and Morphology

Colonial growth of three *M. laxa* isolates was evaluated on PDA-T under different light wavelength ranges in two photoperiods and continuous darkness. After seven days of incubation, growth and conidiation were significantly affected by light conditions and photoperiod (Figure 1) as well as colony morphology (Figure 2). Although the pattern of growth is nearly similar between isolates, there are differences in conidia production in some light conditions among them. Importantly, the temperature was monitored to avoid a cross effect due to illumination energy. The average temperatures were 23 and 23.7 for daylight, 24.5 and 26.5 for cool light, 23.5 and 24.5 for blue-light, 24.1 and 25.9 for green-light, 22.9 and 23.3 for red-light, and 23.5 and 23.9 °C for black light (UV-A), 8 h light and 16 h light photoperiods, respectively.

The most significant growth rate was observed under red light in 5L, 8L, and 25L *M. laxa* isolates, specifically at photoperiod with fewer light hours (8L16D) (Figure 1a–c). All three isolates grew under black light, mainly in the 16L8D photoperiod. At the same light condition, the only significant differences in growth rate were observed between photoperiods in daylight for *M. laxa* 8L and black light for *M. laxa* 25L (Figure 1b,c). None of the three *M. laxa* isolates was able to grow under blue light. In contrast, 8L and 25L grew when they were illuminated with cool white light but at different photoperiods.

The light wavelength and photoperiod influenced the production of conidia in *M. laxa* isolates and was favored with 16L8D photoperiod (Figure 1d–f). In all three *M. laxa* isolates, deficient sporulation was observed in darkness and red light. In the *M. laxa* 5L and 25L, significant differences were observed in daylight, black light, and green light, with the highest production under green light for 5L and black light in 25L. The isolate that best-produced conidia were 8L, especially under black light, followed by daylight or green light. Concerning the morphological colony features, under red light and darkness, the mycelia of the three *M. laxa* were fluffy and off-white. In contrast, concentric rings of sporulation and brownish color were observed in the day, black, and green lights for all three *M. laxa* isolates (Figure 2). Looking at the back of the plates, there are differences in pigmentation between lights, ring-shaped in red light, scattered in darkness, and less pigmentation in daylight, black or green lights (Figure 2).

### 3.2. Identification of Putative Photoreceptors

Based on the described photoreceptors of *B. cinerea* B05.10, we used BLASTP searches, Basic Local Alignment Search Tool (BLAST) for protein searches (P), against the *M. laxa* 8L predicted proteome [32]. As a result, we identified putative orthologous for all the described *B. cinerea* photoreceptors with identities greater than 69% and coverage over 90% except for *velvet4* (Table 1, see also below). All the sequences were verified in a recently *M. laxa* Mlax316 available genome [33] and compared with *M. laxa* 8L (Table 1).

The domain architecture in proteins classified as possible color photoreceptors was as expected according to previous studies in other fungi (Figure 3). The possible red *M. laxa* receptors vary in length between them; all three phytochromes harbor from the N-terminal to C-terminal part a PAS domain (Per/Arnt/SIM, PF08446), GAF domain (cGMP-specific phosphodiesterase, PF01590), PHY domain (phytochrome-specific domain, PF00360), HisK domain (histidine kinase, PF00512), ATPase domain (PF02518), and RRD domain (response regulator domain, PF00072). Putative green light receptors (Figure 3), opsins, harbor a retinal binding site in the rhodopsin domain (PF01036). As blue light photoreceptors, we found the white-collar proteins and the cryptochromes. The white-collar proteins harbor a PAS domain (PF08447) and a Zing finger domain (Zn, PF00320), and MlWC1 had a LOV domain (light–oxygen–voltage) and a different PAS domain (Figure 3). On the other hand, cryptochromes have different domains than white-collar ones; they harbor a PHR domain (Photolyase homology region, PF08005) and a FAD-binding domain (PF03441) (Figure 3).

Due to the lack of a prediction for the velvet 4 coding gene in the published version of the 8L genome [32], we wanted to verify the absence/presence of *velvet4* with specific oligonucleotides to amplify the complete gene of the possible *velvet4* homologous region in *M. laxa* isolates based on the genome sequence of Mlax316 [33]. After performing PCR, obtaining a fragment of the expected size, it was sequenced and proved that this sequence was present in the 8L genome (Appendix A). The prediction of the domain architecture was performed in the same way as the photoreceptors. As expected, MlVEL1, MlVEL3, and MlVEL4 harbor a velvet domain (PF11754), while MlVEL2 had two velvet domains (Figure 3).

### 3.3. Effect of Light on the Transcriptional Pattern of Velvet Gene Family and Identified Photoreceptors

Considering changes in the conidiation, colony morphology, and growth rates when *M. laxa* is exposed to different light sources, expression analysis of putative genes implicated in photoresponses and possible photoreceptors were performed.

The Velvet family has been described in other fungi as involved in photoresponses and included at least four genes in *M. laxa*. Therefore, their transcription pattern was analyzed when *M. laxa* 8L was exposed to white light and blue, green, and red lights, in addition to continuous darkness (Figure 4). Overall, the expression levels of *MlVEL1* were downregulated under any light condition, including continuous darkness, with a strong downregulation with a short daylight exposure (0.5 h). Only green light had a positive effect on *MlVEL1* with a significative peak at 4 h (Figure 4). On the contrary, green light caused a reduction in *MlVEL2* expression than continuous darkness. However, *MlVEL2* expressed higher transcript levels under day, blue, and red lights after two hours of exposure (Figure 4). Expression levels of *MlVEL3* and *MlVEL4* run parallel, increasing after exposure to any light, being the highest levels of expression under green light at 4 h with 6.9–h and 7.1–fold, respectively (Figure 4). Over time there was an increase in expression levels of both genes under continuous darkness.

A subset of representative genes coding for photoreceptors (Figure 3) were also studied under specific illumination conditions. Expression levels of cryptochromes and white-collar genes were studied under blue light, opsins with green light, and phytochromes with red light, in addition to daylight and continuous darkness for all of them.

Perception of blue light can be carried out by two different groups of photoreceptors, white-collar and cryptochrome; the latter also senses in the UV range. Thus, we have analyzed the transcription patterns of two genes of each family under continuous darkness, daylight, and blue light. Cryptochromes, *MlCRY1* and *MlCRY2*, were differentially expressed upon light induction, and the transcription of both genes remained downregulated in constant darkness (Figure 5a). In both genes, *MlCRY1* and *MlCRY2*, daylight positively affected expression peaking at 4 h with 24.5–fold and 17.1–fold, respectively. However, blue light had a more significant effect than daylight in *MlCRY1*, which oscillates in time with immediate effect at 30 min (32.8–fold) and later at 4 h (52.4–fold) (Figure 5a).

On the other hand, we studied the white-collar encoding genes *MlWC1* and *MlWC2* as putative blue light photoreceptors. qRT-PCR results demonstrated a completely different pattern under day or blue light exposure than cryptochromes *MlCRY1* and *MlCRY2* (Figure 5b). Comparing the expression pattern among the three conditions revealed that only blue light triggered a significant upregulation of *MlWC1* at a long exposure of 6 h and *MlWC2* at a short time of 0.5 h. Likewise, both *MlWC1* and *MlWC2* were found significantly repressed in continuous darkness and daylight (Figure 5b).

Regarding putative green light photoreceptors, two opsin encoding genes were analyzed: *MlOPS1* and *MlOPS2* (Figure 6). *MlOPS1* under daylight exposure showed significant upregulation in the interval 0.5–6 h, while a short period of induction was observed under green light exposure (2–4 h). By contrast, *MlOPS1* under continuous darkness showed a short-term downregulation (Figure 6). The expression levels of *MlOPS2* gradually increased under green or daylight with significant upregulation after 2 h of exposure, peaked at 4 h in daylight (23–fold) and green light (8.5–fold), meanwhile remained almost at a basal level compared to control in constant darkness (Figure 6).

In the case of the red light photoreceptors, the phytochrome group showed different expression profiles for each member (Figure 7). The transcript levels of *MlPHY1* and *MlPHY2* showed a slight upregulation in continuous darkness. However, the red light triggered a significantly higher upregulation in *MlPHY1* and daylight in *MlPHY2*. Finally, the expression profile of the last member of the group *MlPHY3* was variable with up- and downregulation waves in continuous darkness. At the same time, red or daylight produced a slight late response (Figure 7).

## 4. Discussion

*Monilinia laxa* is an airborne plant pathogen subjected to different abiotic environmental factors, such as temperature, humidity, wind, and sunlight, that influence brown rot development. Multiple studies have described the importance of temperature and humidity factors for the successful infection in the field [4,37], its influence on the presence of latent infections [2], or even in postharvest fruit management [38,39]. However, few have taken into account the influence of light cycles, quantity, or quality of light in *Monilinia* spp. behavior [5,15,40]. Recently, it has been shown that *M. laxa* is a light-responsive fungal pathogen. Brown rot development is influenced by duration, intensity, and quality of light [16]. Here, we investigated whether *M. laxa* modifies growth and conidiation characteristics in response to different light conditions. In connection, *M. laxa* genome characterization also allowed us to depict potential light receptors and regulatory genes and their transcriptional patterns during light exposure.

The three *M. laxa* strains used in this work have been shown to be photoresponsive. Diverse effects in growth, conidiation, or morphology were observed when changing the photoperiod or light wavelengths. Asexual reproduction is a primary source of *M. laxa* dispersion in Spain [4]. Consequently, knowing the mechanisms mediating the induction of the conidiation process is of importance to understand the spread of brown rot. *M. fructigena* strains have the ability to sense and react to light, e.g., sporulation is more abundant in light–dark cycles than continuous darkness [40]. Therefore, it seems that diurnal patterns trigger conidiation in *M. fructigena* [5] and increases their concentration with greater light exposure in *M. laxa* [16]. The presence of a complete set of velvet genes in the genome of *M. laxa* 8L isolate and in other strains, corroborated by sequencing, predict a fully functional velvet-dependent light-sensing machinery in this fungus. “Blind” strains of closely related *B. cinerea* and *A. nidulans* displaying light-independent conidiation had been shown to carry a mutation in a velvet gene, *BcVEL1* and *veA,* respectively [25,41]. Thus, future searches for blind strains in *M. laxa* envisage the identification of mutations in photoreceptor and light-dependent regulatory systems.

Environmental light condition is a source of information, including signals of space, time, or stress [18]. It is essential to distinguish and anticipate favorable conditions for fungal development [42]. Red light exposure caused the highest growth rate and low conidia production in all three *M. laxa* isolates, especially in the shorter photoperiod, as previously described in *B. cinerea* [13,43]. A significant increase in the brown rot incidence and severity on nectarines caused by *M. laxa* under red light compared to fruit incubated under white light has been identified. Furthermore, similar to the photoresponsive strains of *B. cinerea* [10], *M. laxa* showed deficient conidiation and a high vegetative growth rate under continuous darkness. Such repression in conidia production and mycelial growth promotion could be an adjustment of *M. laxa’s* behavior to signals of low or absence of light [16]. *B. cinerea* achieves its maximum virulence against *Arabidopsis thaliana* at dusk by regulating its circadian machinery [44,45] and when plant defenses against necrotrophic pathogens are lower [46,47]. Promotion of vegetative growth in the dark could also be interpreted as a reflection of the colonization inside the fruit, where no illumination is expected, and *Monilinia* spp. grows vegetatively and only sporulates on fruit surface [1].

In contrast, green and daylight enhanced both growth rate and conidiation in *M. laxa*, showing a higher production of conidia with more hours of exposure to light. The role of green light was less characterized in most fungi [30]; for example, in *N. crassa,* it promotes sexual reproduction [48]. As mentioned above, light is necessary for *B. cinerea* conidiation [10], but the exposure of this pathogen to green light has reversed effects to those described here for *M. laxa*, such as inhibiting germination, growth, and conidiation [14,43]. 

As we observed in *M. laxa*, the promotion of conidiation through the near-ultraviolet light is extended among fungi, for example, *B. cinerea* [13,49], *Alternaria tomato* [50], and *Trichoderma viride* [51], which could be a general reaction to the stress of damaging UV radiation [17,18]. Red and blue lights play an opposite effect during conidiation of *A. alternata* or *B. cinerea* as in other fungi [43,52]. In addition, blue light causes a negative phototropism in the germination of *B. cinerea* [53] and a delay in *A. nidulans* germination [54]. Here we showed a strong inhibitory effect of blue light on plate culture conditions, which was not recorded in *M. laxa*-nectarine infection process [16]; a possible unexplored explanation is that rugosity of fruit surface may contribute to protection to light. Thus, further studies should be done to confirm the possible inhibitory effect of blue light on *M. laxa*. A negative effect had been reported in brown rot development under cool white light that could be due to a combination of high irradiance with temperature [16,55].

The results presented herein demonstrated that *M. laxa* reacts to light–dark cycles and presents diverse behaviors to different light wavelengths, confirming and extending previous reports. Therefore, searches in *M. laxa* 8L genome [32] allowed the identification of velvet regulatory family and photoreceptors already described in the closely related *B. cinerea* [30], except for *MlVEL4*, which had to rely on other *M. laxa* genome for verification [33]. The domain architecture of all the predicted proteins based on the analysis of orthologous genes was expected to be functionally active as photoreceptors. To further investigate the role of the different photoreceptors and the velvet regulatory family, the mycelium samples of our model *M. laxa* 8L were exposed to daylight, different ranges of light wavelengths, and continuous darkness to analyze time-course transcriptional patterns of some of them.

Members of the velvet regulatory family are involved in light-dependent fungal differentiation and have an essential role in coordinating the development and secondary metabolism [27,56]. When analyzing the velvet transcription profile during the light interaction, a general downregulation of *MlVEL1* was observed for both the continuous darkness and different light conditions. The role of the *VEL1* in conidiation has been demonstrated, but its function is dependent on the formation of velvet complex with *VEL2* and the secondary metabolism regulator *LAEA* in both *B. cinerea* and *A. nidulans* [11,57], suggesting that temporal synchronization of these three is necessary for their function. In this sense, the expression of *MlVEL2* was triggered by exposure to blue, red, and daylight, but it also increased in both darkness and green light with time-course.

On the other hand, *MlVEL3* and *MlVEL4* showed higher transcriptional expression under daylight, blue, and specifically green light. Although the role of the latter two is less studied, the fungal development is affected since the absence of *VEL3* in *B. cinerea* causes the production of both conidia and sclerotia in darkness [58]. It would be interesting to investigate the relationships between velvet-complex components and with phytochromes and white-collar proteins as described for *B. cinerea* and *A. nidulans*, among others [11,59]. In addition, the implications of velvet complex in the virulence process of *M. laxa* is important since it could regulate efficient production and secretion of CAZymes and peptidases and subsequently the degradation of host tissues, as shown before in *Aspergillus flavus* and *B. cinerea*, respectively [58,60]. In fact, both virulence-related fungal strategies, tissue acidification, and cell wall degrading enzyme secretion, have been previously described as important factors in *Monilinia* spp.-fruit interaction [61,62].

Blue light sensing is one of the most widespread light senses in filamentous fungi [20]. Since the description of white-collar (*wc1*) as the first related blue light-sensing gene in *N. crassa* [63], blue sensing has been studied in fungal biology [17,21]. No immediate light-triggered effect was observed on the transcripts of *M. laxa* white-collar genes. Similarly, both genes remained practically stable in *B. cinerea* regardless of light exposure or continuous darkness. However, the absence of one affects other photoreceptors, such as opsin and cryptochrome, suggesting regulation by white-collar [10]. In this sense, in *N. crassa,* a primary light effect has been described as the formation of a complex between the proteins derived from both genes, white collar 1 and 2, and subsequently activating the transcription of specific light-responsive genes through direct binding to their promoters [20]. Cryptochromes compose another blue-light-receptor group and are also connected with UV sensing [18,29]. Transcription of *MlCRY1* and *MlCRY2* of *M. laxa* has a strong light-dependence, being upregulated under day and blue lights but downregulated in continuous darkness. In agreement, both cryptochromes are induced by blue or white light in *B. cinerea* in a *WC1* dependent way, although the role of *CRY1* is not clear, and *CRY2* is a negative regulator of conidiation [64].

Opsins are a widespread fungal photoreceptor related to green light-sensing composed of a retinal chromophore, although their functional role remains undetermined [65]. The expression of *M. laxa* opsins was triggered by daylight, especially *MlOPS2*, and slightly in green light. Daylight full light spectrum could have an additive effect in the opsins transcription; it has been described that they can also be activated by blue light [66], although the effect of blue light alone has not been studied here. On the contrary, *N. crassa* opsin1 (*ops-1*) is highly expressed in asexual cultures and is a late-stage conidiation gene, which is under light-independent control of white-collar 2 and related to the developmental stage [67]. Therefore, suggesting a more complex control than direct light activation.

Response to red light does not seem to be so generalized among the filamentous fungi; on the one hand, phytochrome *FphA* of *A. nidulans* acts as a potential red-light sensor and represses sexual development [68]. On the other hand, expression levels of both phytochromes encoded in the *N. crassa* genome are not light-dependent, showing no correlation with exposures to either red, far-red, or daylight exposure [69]. In our study, *M. laxa MlPHY1* and *MlPHY2* phytochromes were red and daylight upregulated, respectively, but light-dependent regulation of *MlPHY3* was imprecise. Similar behavior has been described for *BcPHY2* of *B. cinerea*, which was higher expressed under light exposure than *BcPHY1* or *BcPHY3* [30].

## 5. Conclusions

Taken together, our study provides a general approach to phenotype changes and molecular evidence of transcriptional responses to light in *M. laxa*. The overall picture was that light cycles promote conidiation, but vegetative growth promotion varied under different light conditions. Besides, light transcription patterns support the assumption of *M. laxa* as photoresponsive fungi. Clear responses to light triggered gene expression in *MlVEL2*, *MlPHY2*, *MlOPS2*, and *MlCRY2*, and color light was a positive induction in *MlVEL3*, *MlVEL4*, *MlPHY1*, and *MlCRY1*. The basis for the induction of conidiation or vegetative growth promotion is so far undescribed in *M. laxa*. Further exploration of the contribution of each gene and the transduction pathways involved would be interesting to understand how the adaptative responses take place in *M. laxa*, especially red light photoreceptors, which could be responsible for the increased incidence and severity caused by *M. laxa* on nectarines [16].

## Figures and Tables

**Figure 1 jof-07-00032-f001:**
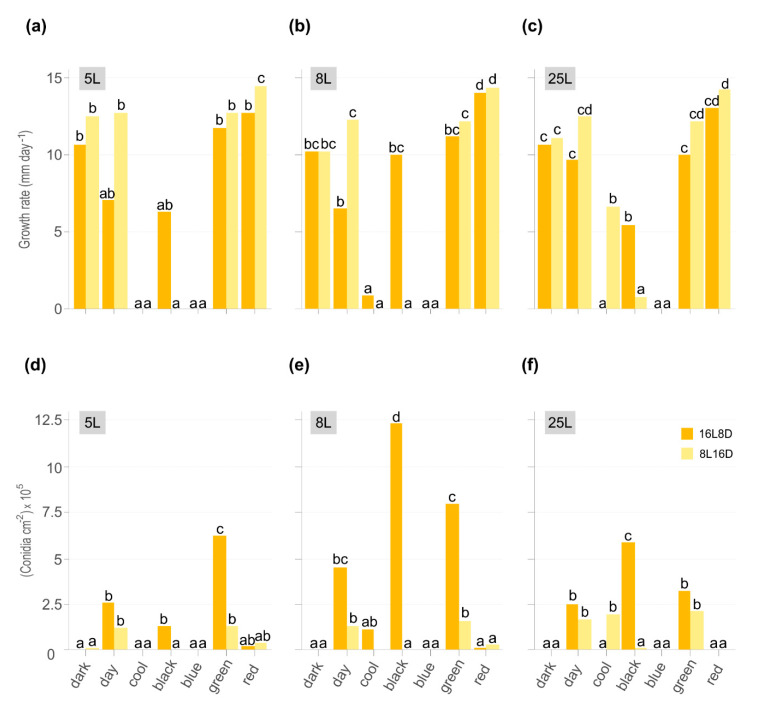
Effect of different light conditions on *Monilinia laxa* growth rate (mm day^−1^) (**a**–**c**) and sporulation (conidia cm^−2^ 10^5^) (**d**–**f**) on PDA-T plates inoculated with three *M. laxa* isolates: 5L (**a**,**d**), 8L (**b**,**e**), and 25L (**c**,**f**). All plates were incubated for 7 days under two white lights (daylight and cool) and black, blue, green, and red lights for photoperiods of 16 h light/8 h darkness (orange) and 8 h light/16 h darkness (yellow). As indicated in materials and methods, the continuous dark condition was subjected to both photoperiods. Data represent the average of two independent experiments, and letters denote significant differences according to the analysis of variance (ANOVA) to Duncan’s multiple range test.

**Figure 2 jof-07-00032-f002:**
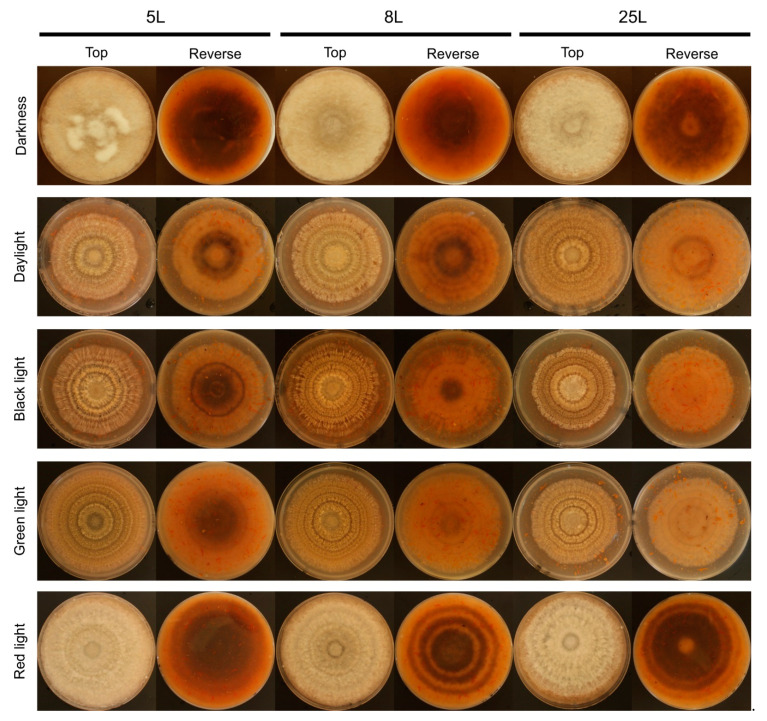
Effect of different light conditions on *Monilinia laxa* colony growth (top and reverse view) on PDA-T plates inoculated with 5L, 8L, and 25L *M. laxa* isolates after 7 days under continuous darkness and illuminated with daylight, black, green, and red lights for 16 h light/8 h darkness.

**Figure 3 jof-07-00032-f003:**
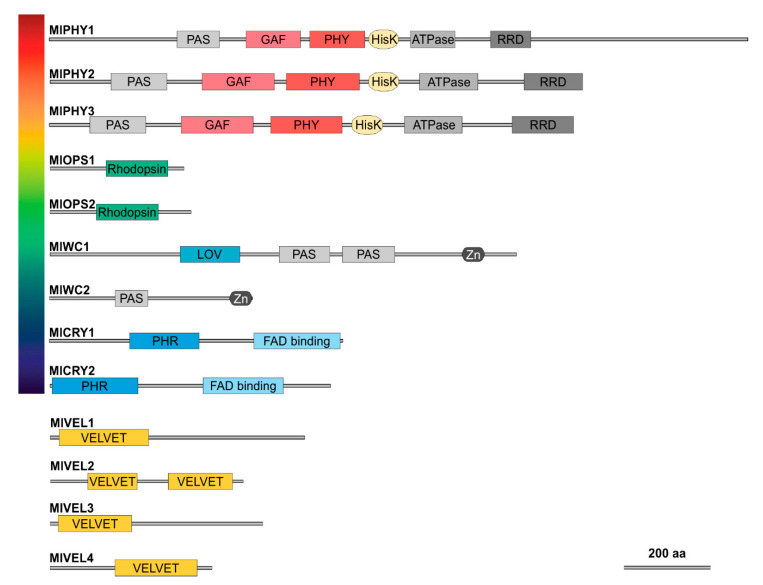
Domain architecture of potential photoreceptors (**top**) and velvet family (**bottom**) proteins in *M. laxa*. PAS (Per/Arnt/SIM, PF08446); GAF (cGMP-specific phosphodiesterase, PF01590); PHY (phytochrome-specific domain, PF00360); HisK (histidine kinase, PF00512), ATPase domain; RRD (response regulator domain, PF00072); rhodopsin domain (PF01036); LOV (light–oxygen–voltage); Zn (Zing finger domain, PF00320); PHR (Photolyase homology region, PF08005); FAD-binding domain (PF03441); and velvet domain (PF11754).

**Figure 4 jof-07-00032-f004:**
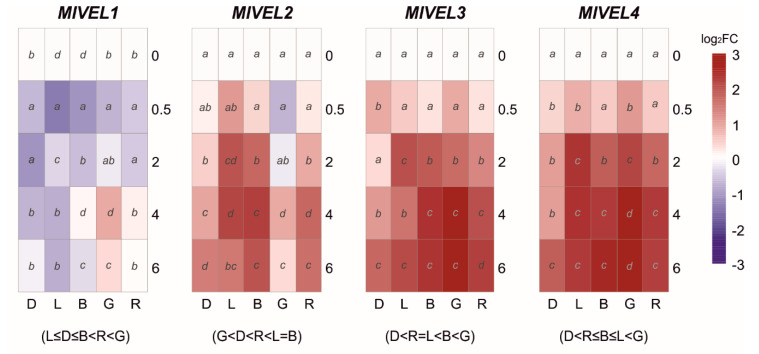
The heat maps represent changes in the relative gene expression of four velvet (*VEL*) coding genes under continuous darkness (D), daylight (L), blue (B), green (G), or red (R) light at different time points. The scale is log_2_ of fold change mean values after normalization against 0 h post illumination using the 2^−ΔΔCT^ method. Red color represents a higher relative expression than 0 h post illumination, and blue color represents a lower relative expression. Data were analyzed between light sources containing all times (shown at the bottom between brackets), and by light source between times (letters) by analysis of variance; the mean values with the same letter are not significantly different (*p* ≤ 0.05) according to Duncan’s multiple range test.

**Figure 5 jof-07-00032-f005:**
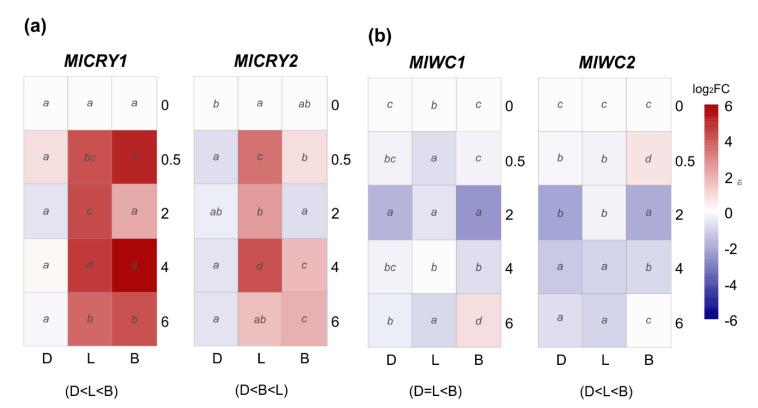
The heat maps represent changes in the relative gene expression of (**a**) two cryptochromes (*CRY*) and (**b**) two white-collar (*WC*) coding genes under continuous darkness (D), daylight (L), or blue (B) light at different time points. The scale is log_2_ of fold change mean values after normalization against 0 h post illumination using the 2^−ΔΔCT^ method. Red color represents a higher relative expression than 0 h post illumination, and blue color represents a lower relative expression. Data were analyzed between light sources containing all times (shown at the bottom between brackets), and by light source between times (letters) by analysis of variance; the mean values with the same letter are not significantly different (*p* ≤ 0.05) according to Duncan’s multiple range tests.

**Figure 6 jof-07-00032-f006:**
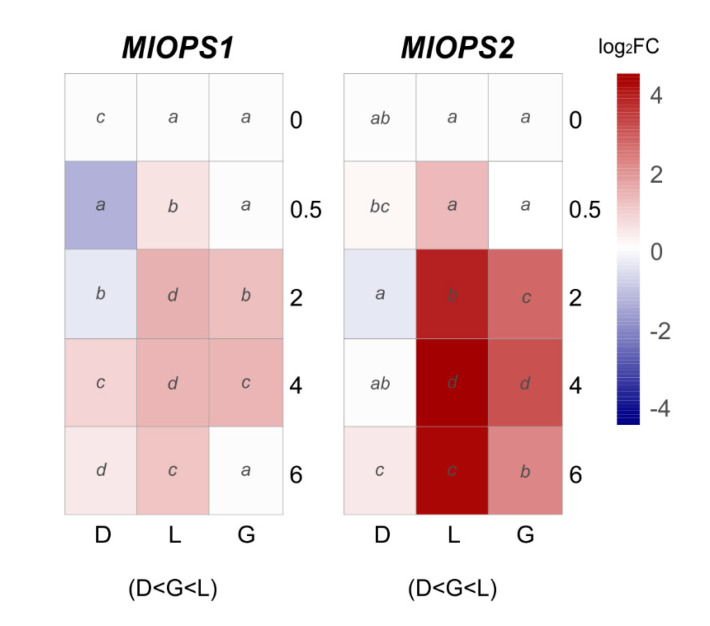
The heat maps represent changes in the relative gene expression of two opsins (*OPS*) coding genes under continuous darkness (D), daylight (L), or green (G) light at different time points. The scale is log_2_ of fold change mean values after normalization against 0 h post illumination using the 2^−ΔΔCT^ method. Red color represents a higher relative expression than 0 h post illumination, and blue color represents a lower relative expression. Data were analyzed between light sources containing all times (shown at the bottom between brackets), and by light source between times (letters) by analysis of variance; the mean values with the same letter are not significantly different *p* ≤ 0.05) according to Duncan’s multiple range test.

**Figure 7 jof-07-00032-f007:**
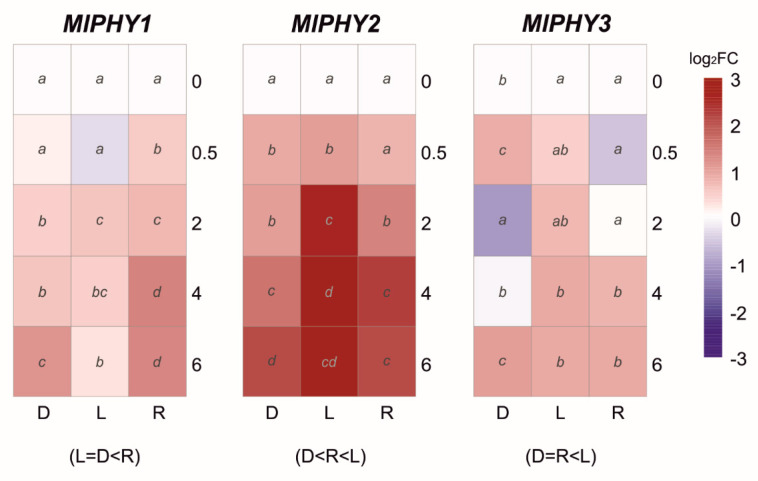
The heat maps represent changes in the relative gene expression of three phytochromes (*PHY*) coding genes under continuous darkness (D), daylight (L), or red (R) light at different time points. The scale is log_2_ of fold change mean values after normalization against 0 h post illumination using the 2^−ΔΔCT^ method. Red color represents a higher relative expression than 0 h post illumination, and blue color represents a lower relative expression. Data were analyzed between light sources containing all times (shown at the bottom between brackets), and by light source between times (letters) by analysis of variance; the mean values with the same letter are not significantly different (*p* ≤ 0.05) according to Duncan’s multiple range test.

**Table 1 jof-07-00032-t001:** Proteins implicated in photoresponses and potential photoreceptors of *Monilinia laxa* 8L.

Gene Name	Gene/Protein ID ^a^	GenBank Accession	Putative Protein Function ^b^	Predicted Protein Length ^c^	*B. cinerea* B05.10 ^d^	% Identity	% Coverage	*M. laxa* Mlax316 ^e^	% Identity	% Coverage
**Putative near-UV/blue light sensors**										
*MlCRY1*	Monilinia__090370	BK014380	putative deoxyribodipyrimidine photo-lyase	679	Bcin05g08060	77.80	90.27	EYC80_003544	99.68	100
*MlCRY2*	Monilinia__028340	BK014381	DASH family cryptochrome protein	649	Bcin09g01620	86.59	95.13	EYC80_007030	99.82	91.84
**Blue light sensing**										
*MlWC1*	Monilinia__016540	BK014382	putative white collar-1 protein	1094	Bcin02g07400	69.52	93.61	EYC80_007313	99.50	100 *
*MlWC2*	Monilinia__092390	BK014383	putative white collar-2 protein	483	Bcin05g05530	72.94	91.36	EYC80_003689	100	100
*MlVVD1*	Monilinia__019850	BK014384	vivid PAS VVD protein	237	Bcin13g01270	83.10	96.98	EYC80_009309	100	100
*MlLOV3*	Monilinia__014120	BK014393	similar to Bclov3	836	Bcin10g03870	78.98	96.50	EYC80_002606	99.88	100
*MlLOV4*	Monilinia__012050	BK014394	regulator of G protein	588	Bcin02g04390	79.93	98.65	EYC80_003150	100	100
**Green light sensing**										
*MlOPS1*	Monilinia__070650	BK014385	putative opsin-1 protein	310	Bcin02g02670	85.26	99.36	EYC80_005142	100	100
*MlOPS2*	Monilinia__055890	BK014386	putative opsin-like protein	327	Bcin01g04540	79.35	96.45	EYC80_010258	100	100
**Red light sensing**										
*MlPHY1*	Monilinia__010410	BK014390	PHY1, histidine kinase-group VIII protein	1616	Bcin13g04690	79.18	98.71	EYC80_009624	99.94	100
*MlPHY2*	Monilinia__052440	BK014391	PHY2, histidine kinase-group VIII protein	1222	Bcin01g09230	75.24	99.51	EYC80_005619	96.97	100
*MlPHY3*	Monilinia__088010	BK014392	PHY3, histidine kinase-group VIII protein	1196	Bcin06g01290	79.24	98.84	EYC80_000954	98.16	96.46
**Proteins implicated in photoresponses**										
*MlVEL1*	Monilinia__005480	BK014387	velvet complex subunit 1	589	Bcin15g03390	75.17	99.48	EYC80_008048	99.66	95.77
*MlVEL2*	Monilinia__093380	BK014388	velvet complex subunit 2	446	Bcin01g02730	89.02	93.79	EYC80_005890	100	100
*MlVEL3*	Monilinia__072950	BK014389	velvet 3	492	Bcin03g06410	78.64	98.00	EYC80_000111	100	100
*MlVEL4*	Monilinia__100000	MW349131	velvet 4	375	Bcin07g05880	92.70	95.41	EYC80_003351	98.70	95.66
**Proteins implicated in the circadian clock**										
*MlFRQ1*	Monilinia__081690	BK014395	putative frequency clock protein	968	Bcin02g08360	71.25	98.55	EYC80_007367	97.93	100

^a^ Gene/protein unique identifier in 8L *M. laxa* genome. ^b^ Putative protein function based on functional annotation. ^c^ Predicted protein length in 8L *M. laxa* proteome. ^d^ Gene unique identifier in *Botrytis cinerea* genome; BLASTP coverage and BLASTP identity. ^e^ Gene unique identifier in Mlax316 *M. laxa* genome; BLASTP coverage and BLASTP identity. * *M. laxa* entry EYC80_007313 (705 aa) partial align with Monilinia__016540 (1094 aa).

## Data Availability

The data presented in this study are available in this article and Appendix A.

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
