# Peer review of "Light-Photoreceptors and Proteins Related to Monilinia laxa Photoresponses"

_jof, 2021, doi:10.3390/jof7010032_

Round 1
Reviewer 1 Report
Silvia Rodríguez-Pires and colleagues present an article on the in silico identification of the putative light-sensing machinery of the phytopathogenic fungus Monilinia laxa, focusing on growth phenotypes in specific conditions and the transcriptional response of specific genes. In general the article contains interesting and important information for people in the field, is comprehensibly written and fits within the scope of the journal.
Authors are mainly dealing with primary effects of specific wavelengths on aspects of the asexual cycle, or vegetative growth. As stated in the Introduction, the activity of photoreceptors influences multiple biological responses such as spore germination, vegetative growth, sexual and asexual development, secondary metabolism, and circadian rhythm. It would be most informative, and in my opinion also necessary, if more data are given in relation to other stages of the life cycle of M. laxa, like for example spore germination.
Authors should carefully edit the manuscript, in order to to avoid common errors in grammar or syntax. A few examples are given below:
Line 46: please change “factor” to “factors”.
Line 77: please change “light-responsive fungi” to “a light-responsive fungus”.
Lines 69-70, 339-340, 344-348, 406-408: please rephrase.
Line 226: Please explain why the identification of Vel4 was so important.
Author Response
Referee 1: I would like to thank the referee for her/his general comments on the manuscript
Specific comments:
- It would be most informative, and in my opinion also necessary, if more data are given in relation to other stages of the life cycle of M. laxa, like for example spore germination.
- We have not analyzed the speed of germination under different lighting conditions; if we consider that it would be important in future work to establish from which point in time of germination and mycelial growth light takes on an important role through the expression of the different photoreceptors. Since Monilinia is a pathogen that grows mainly as a vegetative mycelium inside the fruit, we have focused on understanding the regulation of the different photoreceptors in these undifferentiated cells.
- Line 46: please change “factor” to “factors”
- It has been done (line 44) in new version
- Line 77: please change “light-responsive fungi” to “a light-responsive fungus”.
- It has been done (line 80) in new version
- Please rephrase Lines 69-70.
- It has been done (lines 70-72) in new version
- Please rephrase Lines 339-340.
- It has been done (lines 347-349) in new version
- Please rephrase Lines 344-348.
- It has been done (lines 357-361) in new version
- Please rephrase Lines 406-408.
- It has been done (lines 422-424) in new version
- Line 226: Please explain why the identification of Vel4 was so important.
- Perhaps we have over emphasized the finding of VEL4 in 8L strain. We wanted to note that the velvet family is complete in 8L strain since initial predictions using Augustus did not render any candidate for a gene coding VEL4 protein. Notably direct PCR sequencing confirmed the presence of a vel4 gene, although two extra amino acids are predicted in VEL4 protein in comparison with other Monilinia laxa. We have reviewed the section where this gene is described (lines 230-231) in new version.

Reviewer 2 Report
The European brown rot Monilinia laxa (Aderhold et Rulhand) is a very big issue in agriculture and causes diseases in important Rosaceae family crops in particular stone fruit and pome fruit
Authors have full legitimity to conduct such a study, and already published very nice works on European brown rot Monilinia laxa:
Depicting the battle between nectarine and
Monilinia laxa: the fruit developmental stage
dictates the effectiveness of the host defenses and the pathogen’s infection strategies
Marta Balsells-Llauradó1, Christian J. Silva2, Josep Usall1, Núria Vall-llaura1, Sandra Serrano-Prieto1, Neus Teixidó1, Saskia D. Mesquida-Pesci2, Antonieta de Cal3, Barbara Blanco-Ulate2 and Rosario Torres1
Many abiotic environmental factors have been studied and should be deeper investigated
light is among them
one point should be extended, for non-specialists
close taxonomic relationship between Botrytis cinerea & Monilinia laxa
lines 74-75 The closely related B. cinerea possesses a remarkable collection of photoreceptor genes covering from UV to far-red light [30].
lines 48-50 Light responsive Botrytis cinerea strains through sensing specific ranges of light wavelength show different colonial growth behaviours.
= if they are so close, similar photoresponse system expected?
lines 80-83
The main objective of this work was to evaluate the effect of light on colony growth and conidiation of Monilinia laxa in vitro conditions and describe the putative photoreceptor machinery in M. laxa genome by in silico analyses. The transcriptional pattern of some potential photoreceptor genes in response to different light wavelengths was also determined.
all objectives presented in the introduction are justified and then are well conducted by the experiments, data and interpretation provided
as plant cultures attacked by Monilinia laxa are orchards located outside, open air, open daylight what would be the strategies to limit the development of this fungi, based on the new data provided in this paper
what about strategies conducted in other plant pathogens with a similar photoresponse system?
Is Monilinia laxa also involved in postharvest fruit diseases?
Author Response
Referee 2: I would like to thank the referee for her/his general comments on the manuscript
Specific comments:
- one point should be extended, for non-specialists close taxonomic relationship between Botrytis cinerea & Monilinia laxa.
- It has been done (line 76) in new version
- lines 74-75 The closely related B. cinerea possesses a remarkable collection of photoreceptor genes covering from UV to far-red light [30].lines 48-50 Light responsive Botrytis cinerea strains through sensing specific ranges of light wavelength show different colonial growth behaviours.= if they are so close, similar photoresponse system expected?3. as plant cultures attacked by Monilinia laxa are orchards located outside, open air, open daylight what would be the strategies to limit the development of this fungi, based on the new data provided in this paper4. what about strategies conducted in other plant pathogens with a similar photoresponse system? 5. Is Monilinia laxa also involved in postharvest fruit diseases?
- Yes it is. Please see page 1 line39
- Citation to previous work done in B. cinerea is present along the manuscript. If these question relates to previous comment is out of our knowledge whether light is currently used as a way of control of this disease.
- Based on this work and in previous work (Rodríguez-Pires et al., 2020), we conclude that certain wavelengths would be useful to reduce brown rot at postharvest. At the moment we have no developed this option with packinghouse fruit sector, and looked to us premature proposing in our conclusions immediate application of these results in the sector.
- In fact, the identification of putative genes coding photoreceptors and other proteins involved in light sensing is based on conservation of sequences among fungal kingdom. Thus, we expected conservation of elements in these photoresponsive systems, but probably how signaling occurs or target genes would differ among species. These differences are in our aim to find, that could explain differential behaviors of these pathogens, up to date we have mainly focus on general analysis of machinery for sensing light because this topic has not been approached in our model fungus. Future studies will shed light on these differences and their roles in pathogenicity of Monilinia spp compared to other members of Sclerotiniaceae family.

Round 2
Reviewer 1 Report
Although I still think that it would be beneficial to include a few more data on the expression of such genes during other developmental stages, I realize that authors do not wish to engage in further experiments and prefer to deal with such aspects in a future study.
I have no further comments for the authors.